# The Molecular Genetic Epidemiology and Antimicrobial Resistance of *Neisseria gonorrhoeae* Strains Obtained from Clinical Isolates in Central Panama

**DOI:** 10.3390/microorganisms11102572

**Published:** 2023-10-16

**Authors:** Virginia Núñez-Samudio, Mellissa Herrera, Genarino Herrera, Gumercindo Pimentel-Peralta, Iván Landires

**Affiliations:** 1Instituto de Ciencias Médicas, Las Tablas 0710, Los Santos, Panama; gumercindopimentel@gmail.com; 2Sección de Epidemiología, Departamento de Salud Pública, Región de Salud de Herrera, Ministry of Health, Chitré 0601, Herrera, Panama; 3Laboratorio Clínico, Hospital Luis “Chicho” Fábrega, Región de Salud Veraguas, Ministry of Health, Santiago 0923, Veraguas, Panama; myherrera@minsa.gob.pa (M.H.); ghc_22@hotmail.com (G.H.); 4Hospital Regional Dr. Joaquín Pablo Franco Sayas, Región de Salud de Los Santos, Ministry of Health, Las Tablas 0710, Los Santos, Panama

**Keywords:** gonorrhea, resistance, antibiotics, molecular epidemiology, genetics

## Abstract

We aim to analyze *Neisseria gonorrhoeae* isolates in central Panama, characterize the associated gonococcal antimicrobial resistance (AMR) and conduct molecular epidemiology and genetic typing. We conducted a retrospective study based on *N. gonorrhoeae* hospital isolates collected between 2013 and 2018. AMR was determined using dilution agar and Etest^®^. Molecular typing was conducted using the Multilocus Sequence Typing (ST) scheme. The isolates analyzed (*n* = 30) showed resistance to penicillin (38%), tetracycline (40%), and ciprofloxacin (30%), and sensitivity to extended-spectrum cephalosporins and azithromycin. We identified 11 STs, the most frequent of which was ST1901 among the strains with decreased sensitivity and resistance to three types of antibiotics. We identified eight variations for the *penA* gene, all non-mosaic, with type II LVG as the most frequent (50%). To the best of our knowledge, we conducted the first Central American genomic study that analyzes a collection of gonococcal isolates, which represents a benchmark for future epidemiological and molecular genetic studies. The high prevalence of ciprofloxacin, tetracycline, and penicillin resistance, in addition to the identification of the worldwide spread of multidrug-resistant clone ST1901, should prompt the continuous and reinforced surveillance of AMR, including the molecular epidemiology of *N. gonorrhoeae* in Panama.

## 1. Introduction

*Neisseria gonorrhoeae* is a pathogen exclusive to humans, which is responsible for gonorrhea, one of the most prevalent sexually transmitted infections (STIs) worldwide. In men, it usually manifests as urethritis and epididymoorchitis, while women usually develop asymptomatic cervicitis [1]. According to estimates from the World Health Organization (WHO), in 2020, 82.4 million people were infected with *N. gonorrhoeae* [2]. The global incidence rate (IR) estimate for the same year was 1124 per 100,000 inhabitants, showing an upward trend compared to 2017. In Latin America, the IR estimates vary by region, observing 255 per 100,000 inhabitants in the Andean region and 1199 per 100,000 inhabitants in the Caribbean region [3].

*N. gonorrhoeae* has increased its antimicrobial resistance (AMR) to available antibiotic treatments (i.e., penicillins, tetracyclines, ciprofloxacin) [4]. Currently, extended-spectrum cephalosporins (ESCs) are the last line of available antibiotics recommended for the treatment of gonococcal infection, either as a single dose of ceftriaxone [5] or in combination therapy with azithromycin [6]. In 2017, *N. gonorrhoeae* was cataloged by the WHO among the pathogens resistant to antibiotics with priority for research purposes [7]. It is urgent to prevent and control AMR so that gonorrhea remains a treatable infection. However, increasing resistance to azithromycin and ceftriaxone has also been reported. Global data from the WHO global antimicrobial resistance surveillance for *N. gonorrhoeae* for 2017–2018 show variability in the annual proportion of ceftriaxone (0–21%) and azithromycin (0–60%) resistance [8]. In Latin America, an increase in the annual proportion of resistance to ceftriaxone between 0% and 4.9% was registered for the same period and between 0% and 29.9% for azithromycin [8,9]. Countries such as Argentina and Brazil have described strains with resistance to azithromycin and ceftriaxone [10,11,12] present mainly in ST1901. In turn, this sequence type (ST) harbors the mosaic allele *penA*-34, which belongs to the worldwide spread of multidrug-resistant clones NG-MAST ST1407 and NG-STAR ST90, which are both associated with a reduced susceptibility to ESCs in many countries [10,11].

β-lactam antibiotics have been widely prescribed for the treatment of gonorrhea. One of the main resistance mechanisms among *N. gonorrhoeae* strains is the alteration of penicillin-binding protein 2 (PBP2), which is encoded by the *penA* gene [4]. Mutations in the *penA* gene lead to a reduction in sensitivity to these antibiotics due to the modification in their target of action. Currently, some 489 alleles of the PBP2 protein have been described, and are divided into 234 main types. However, reduced susceptibility or resistance to ESCs is usually associated with mosaic *penA* alleles, that is, with multiple simultaneous polymorphisms [13]. Another resistance mechanism to β-lactams is the plasmid-mediated acquisition of TEM-type β-lactamase (penicillinase) enzymes [14]. On the other hand, the mechanisms of resistance to quinolones have been described for *N. gonorrhoeae* through the chromosomal amino acid substitutions of the *gyrA* and *parC* genes (the targets of quinolones), while for azithromycin, mutations have been described in the genes that code for quinolones: the 23S rRNA subunit leading to its resistance [15].

Genomic studies using molecular epidemiology are paramount to understand the gonococcal population as it evolves and, thus, determine the dynamic changes in AMR patterns. However, in the Central American region, these types of studies are scarce. The objective of this work was to analyze the strains of *N. gonorrhoeae* isolated in a clinical laboratory of a reference hospital in the central region of Panama, characterize the associated gonococcal AMR, and conduct typing through molecular epidemiology and genetic studies, with the aim of knowing the genotype of these samples and detecting epidemiologic clusters and their susceptibility patterns.

## 2. Materials and Methods

### 2.1. Study Design

We conducted a retrospective epidemiological study from a collection of 30 isolates of *N. gonorrhoeae* collected as part of routine clinical care and laboratory surveillance during 2013–2018 at the Dr. Luis “Chicho” Fábrega Hospital. This hospital is one of the main centers providing medical care and clinical laboratory services in the central region of Panama.

A technical sheet was completed anonymously for each sample collected, recording the following patient data: sex, age, and anatomical site. The recorded data were captured in MS Excel (The Microsoft Corporation; Redmond, WA, USA). Data analyses were performed using Stata 11.0 statistical analysis software (StataCorp, LLC; College Station, TX, USA). We calculated descriptive statistics and used Fisher’s exact test to compare proportions and the Mann–Whitney U test to compare medians, setting alpha at 0.05 for statistical significance.

### 2.2. Antibiotic Susceptibility Test

The isolates were recovered on chocolate agar (BioMérieux; Marcy l’Etoile, France). Catalase and oxidase tests and beta-lactamase detection (nitrocefin disks) were performed. Antibiotic susceptibility was determined using the disk diffusion method on GC base agar under the quality standards established by the WHO’s Gonococcal Antimicrobial Surveillance Programme (GASP). Antibiotic susceptibility tests were performed on each of the isolates from the collection and interpreted according to the M100 Performance Standards for Antimicrobial Susceptibility Testing [16]. Sensitivity tests were conducted for antibiotics recommended by the GASP. The following antibiotic discs were used: azithromycin (AZM 15 µg), cefixime (CFM 30 µg), ceftriaxone (CRO 30 µg), ciprofloxacin (CIP 5 µg), penicillin (PEN, 30 µg), and tetracycline (TET 30 µg). The data obtained were classified as resistant, intermediate, and sensitive. The Minimum Inhibitory Concentration (MIC) test was performed using the Etest^®^ technique (BioMérieux; Marcy l’Etoile, France) in isolates resistant to antibiotics, as established in the GASP. Inhibition halos and MICs were recorded in the WHOnet database.

### 2.3. Molecular Typing Analysis and Molecular Identification of β-Lactamase

Molecular typing analyses were performed using the Multilocus Sequence Typing (MLST) scheme for a *N. gonorrhoeae*-standardized protocol [17]. Internal sequencing fragments of seven genes (i.e., *abcZ*, *adk*, *aroE*, *fumC, gdh*, *pdhC*, and *pgm*) were amplified from the chromosomal DNA of the *N. gonorrhoeae* isolate. The sequencing of polymerase chain reaction (PCR) products was performed using the services of Macrogen (Macrogen Inc.; Seoul, Korea). Gene sequences were analyzed using Geneious prime v. 2020.5 (Biomatters, Ltd.; Auckland, New Zealand) and the allelic profile was determined using *N. gonorrhoeae* MLST databases (https://pubmlst.org/organisms/neisseria-spp; accessed on 8 July 2023).

All isolates with β-lactam resistance phenotypes were analyzed for *blaTEM* and *penA* genes using specific PCR primers: TEM-F (5′GCGGAACCCCTATTTG), TEM-R (5′ACCAATGCTTAATCAGTGAG), *penA*-F (5′ATCGAACAGGCGACGATGTC), and penA-R (5′GATTAAGACGGTGTTTTGACGG), as previously described [18,19]. The amplified samples were sequenced and analyzed in the BLASTN program of the National Center for Biotechnology Information (https://blast.ncbi.nlm.nih.gov/Blast.cgi; accessed on 10 July 2023). The alignment and multiple assembly of *penA* were carried out with partial sequences (https://www.ncbi.nlm.nih.gov/nuccore/M32091; accessed on 10 July 2023).

## 3. Results

A total of 30 isolates of *N. gonorrhoeae* collected from clinical samples during the years from 2013 to 2018 were analyzed in this study; 80% (24/30) of the samples came from urethral secretion, 13% (4/30) vaginal, and 7% (2/30) ocular. In total, 87% of the patients were male and 13% were female (*p* < 0.001), with a median (IQR) age of 21 (17.5–24.5) years.

Figure 1 shows the percentage of resistance to antibiotics in the *N. gonorrhoeae* strains analyzed in this study. We observed that 38% of the isolates (11/30) were resistant and 60% (18/30) presented intermediate sensitivity to penicillin. Regarding ciprofloxacin, 30% (9/30) were resistant and 10% (3/30) registered intermediate sensitivity; 40% (12/30) were resistant to tetracycline and 43% (13/30) showed intermediate sensitivity to this antibiotic. All isolates were sensitive to cefixime, ceftriaxone, and azithromycin. In general, among the antibiotics analyzed, the percentage of nonsensitive (i.e., resistant + intermediate sensitivity) was 97% (29/30) for penicillin (*p* < 0.001), 83% (25/30) for tetracycline (*p* < 0.0003), and 40% (12/30) for ciprofloxacin (*p* < 0.273).

Table 1 shows the phenotypic and genotypic profile of the *N. gonorrhoeae* isolates. We observed that, in 55% (16/29) of the strains not sensitive to penicillin, the β-lactamase test was positive, identifying the *blaTEM* gene in 81% (13/16) (*p* = 0.0001).

Using the MLST molecular typing technique, 11 STs were identified in the *N. gonorrhoeae* samples analyzed, which are described in Table 1. The most frequent STs identified were ST11516 (20%) and ST8145 (13%). STs 1584, 1901, and 1893 were identified in 10% of the isolates. In four isolates, it was not possible to determine the STs due to their incomplete allelic profile. Among the strains resistant to penicillin, ST8145 was the most frequent 4/11, while among the strains with intermediate sensitivity to penicillin, ST11516 (4/18), ST1901 (3/18), and ST1893 (3/18) were the most frequent. ST1901 was the most prevalent among the strains with resistance and intermediate susceptibility to three types of antibiotics (ciprofloxacin, tetracycline, and penicillin).

Regarding the *penA* gene patterns, the nucleotide analysis determined a total of eight amino acid sequence patterns (Table 1). Five of the eight variations that we observed have been previously described [14,21]. The sequence of pattern II (LVG) was found in 50% (15/30) and pattern XIV (LVGN) in 23% (7/30), representing 73% of the total isolates analyzed. Among the strains with resistance to penicillin, the most frequent pattern was XIV (6/11), while for those with intermediate sensitivity it was II (11/18).

## 4. Discussion

In Central America, studies on gonococcal AMR are very limited, particularly data referring to molecular and genetic epidemiology. In this study, the isolates of *N. gonorrhoeae* collected from clinical samples from 2013 to 2018 in a hospital in the central region of Panama were analyzed. To the best of our knowledge, this is the first genomic study in Central America and Panama that analyzes a collection of gonococcal isolates, which represents a benchmark for future molecular epidemiological studies.

Of the total *N. gonorrhoeae* strains analyzed, we found a resistance of 38%, 40%, and 30% for penicillin, tetracycline, and ciprofloxacin, respectively, which are results that generally coincide with information obtained from other regions of the world [8]. In Latin America, for example, a high resistance of *N. gonorrhoeae* to antibiotics such as penicillin has been described [22] which oscillates between 17.6% and 98%, to tetracycline of between 20.7% and 90%, and between 5.9% and 89% to ciprofloxacin. The results of this study show that resistance to these three types of antibiotics has been present in all the years of isolation of the strains (2013–2018). Among the isolates analyzed, no resistance to azithromycin or ESCs was reported; however, in some countries of the region such as Argentina, Brazil, and Peru, strains of *N. gonorrhoeae* resistant to azithromycin [14] and ESCs (cefixime and ceftriaxone) have been identified [22,23,24]. It has also been highlighted that surveillance data for gonorrhea are nonexistent or very limited in some Latin American countries [25], which represents a great challenge for its surveillance.

In the Americas, the prevalence of STs obtained through the MLST technique from *N. gonorrhoeae* isolates may vary. For example, in the United States, the STs 9363,10314, 8143, and 1599 have been identified as the predominant ones [26], while in Brazil, the STs 1901 and 1588 have been reported [27]. Using the MLST molecular typing technique, this study identified 11 STs (Table 1), which shows the existence of significant circulating genetic diversity. STs 11516 and 8145 were the most predominant in this study, which does not match the prevalence of the STs described for the US and Brazil. On the other hand, we identified ST8145 in strains analyzed in Brazil, with an AMR pattern that differs from those strains isolated in Panama, where ST8145 was resistant to PEN and sensitive to CIP, while those isolates from Brazil were all resistant to CIP [27].

ST1901 was the most frequent among *N. gonorrhoeae* strains with decreased sensitivity and resistance to three types of antibiotics: penicillin, ciprofloxacin, and tetracycline. Nonmosaic *penA* alleles II and IV were identified in ST1901, which are associated with resistance to penicillin and ciprofloxacin, but are not associated with resistance to ESCs. Published data show a global decrease in the frequency of this ST1901 in strains of *N. gonorrhoeae* analyzed in 2018 [28,29]. In our region, molecular genetic studies carried out on strains with resistance to ESCs in Brazil and Argentina have mainly identified ST1901 which carries the *penA*-34 mosaic allele belonging to the worldwide spread of the multidrug-resistant clone NG-MAST ST1407 and NG-STAR ST90, which are significantly associated with resistance to ESCs [8,10,11]. Bacterial evolution studies suggest that *penA*-34 in ST1901 was generated from *penA*-10 through recombination with another *Neisseria* species, followed by recombination with a gonococcal strain harboring wild-type *penA*-1 [30].

Our data showed non-mosaic type *penA* alleles in all the samples analyzed with intermediate sensitivity or resistance to penicillin, with the type II LVG pattern being the most prevalent (50%). It has been described that the greatest allelic diversity of *penA* is observed in pattern II, being the most common type of non-mosaic allele [13]. Reduced susceptibility or resistance to ESCs is usually associated with mosaic *penA* alleles, mostly *penA*-34. In our study, non-mosaic or semi-mosaic alleles were identified in the analyzed strains. However, the identification of eight amino acid sequence patterns among the PBP2 proteins in this study indicates a considerable presence of variation, which determines the importance of continuously monitoring STs, such as ST1901, which has enormous public health importance due to its capacity for genetic combination that can provide greater AMR [30].

In Panama, the national guidelines for a comprehensive approach to STIs [31] recommend ceftriaxone monotherapy as the first-line treatment for gonorrhea and single-dose ciprofloxacin for patients allergic to penicillins. For the management of urethral discharge syndrome, the guidelines recommend ceftriaxone or ciprofloxacin in a single dose, plus doxycycline for 7 days or azithromycin in a single dose (for the treatment of *Chlamydia trachomatis* and gonorrhea). This study showed a high resistance to ciprofloxacin (30%) among the analyzed strains; a result that coincides with the information obtained from other regions of Latin America [22], so these findings should be considered in the empirical STI treatment guidelines in Panama [31].

As a limitation of the present study, we mention that it was very difficult to obtain clinical samples due, in the first place, to the low number of requests for cultures by the attending physicians; thus, the results must be interpreted with caution given the limited sample size.

Despite its limitations, this study makes unprecedented contributions to the knowledge of the AMR and molecular epidemiology of *N. gonorrhoeae* strains in Panama and Central America with the identification of different clones, including ST1901 which is widely disseminated worldwide. It has a high capacity to evolve to higher levels of AMR, which calls our attention to potential future difficulties in the treatment of gonorrhea infections, as well as the importance of knowing the composition and distribution of resistance genotypes to antibiotics as an important step to establish public policies aimed at delimiting the impact of infections of *N. gonorrhoeae* in Panama and the Central American region.

## Figures and Tables

**Figure 1 microorganisms-11-02572-f001:**
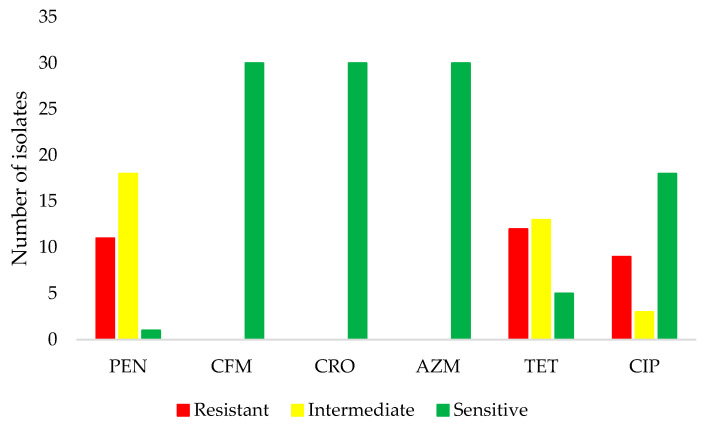
Number of *Neisseria gonorrhoeae* isolates not sensitive to antibiotics (*n* = 30). Abbreviations: AZM, azitromycin; CFM, cefixime; CIP, ciprofloxacin; CRO, ceftriaxone; PEN, penicillin; TET, tetracycline.

**Table 1 microorganisms-11-02572-t001:** Genotypic and phenotypic characteristics of *Neisseria gonorrhoeae* isolates.

Isolate	ST MLST	Year	Origin	PEN	CFM	CRO	AZM	TET	CIP	β-Lactamase/Test	Substitution *	*penA* Pattern
1	11516	2017	V	I	S	S	S	I	S	−	LVG	II
2	1584	2017	O	R	S	S	S	I	S	TEM/+	LVGN	XIV
3	1893	2017	U	I	S	S	S	I	R	−	LVG	II
4	11516	2018	U	I	S	S	S	S	S	−	VLVGS	VII
5	11516	2017	V	I	S	S	S	S	S	−	LVG	II
6	10932	2018	U	I	S	S	S	I	S	−	LVG	II
7	8145	2016	O	R	S	S	S	I	S	TEM/+	LVG	II
8	8145	2016	V	R	S	S	S	I	S	TEM/+	LVG	II
9	ND	2017	U	I	S	S	S	R	I	−	LVG	II
10	11516	2017	U	I	S	S	S	I	S	−	LVG	II
11	1901	2017	U	I	S	S	S	R	R	TEM/+	LVGS	IV
12	1893	2015	U	I	S	S	S	R	S	TEM/+	LVG	II
13	ND	2017	U	I	S	S	S	I	R	+	LVGN	XIV
14	8145	2017	U	R	S	S	S	R	S	TEM/+	LVGN	XIV
15	ND	2014	U	R	S	S	S	R	R	TEM/+	LVGNVNV	XIX
16	1901	2016	U	I	S	S	S	R	R	TEM/+	LVGS	IV
17	1893	2014	U	I	S	S	S	R	I	TEM/+	LVG	II
18	11148	2018	U	I	S	S	S	R	S	−	LVG	II
19	7367	2018	U	I	S	S	S	I	I	−	N	XV
20	7367	2016	U	I	S	S	S	R	R	−	TLVG	44 **
21	8145	2018	U	R	S	S	S	I	S	TEM/+	LVGN	XIV
22	1584	2017	V	R	S	S	S	R	R	+	LVGN	XIV
23	11516	2018	U	S	S	S	S	S	S	−	LVG	II
24	ND	2017	U	R	S	S	S	I	R	TEM/+	LVGN	XIV
25	1901	2017	U	I	S	S	S	R	R	TEM/−	LVG	II
26	1902	2013	U	R	S	S	S	I	S	TEM/+	LVGS	IV
27	10932	2015	U	I	S	S	S	R	S	−	LVGNVQVVNV	XXII
28	1905	2014	U	I	S	S	S	I	S	−	LVG	II
29	1584	2013	U	R	S	S	S	S	S	TEM/+	LVGN	XIV
30	11516	2014	U	R	S	S	S	S	S	+	LVG	II

Abbreviations: AZM, azithromycin; CIP, ciprofloxacin (S ≥ 41 mm); CFM, cefixime (S ≥ 31 mm); CRO, ceftriaxone (S ≥ 35 mm); I, intermediate; MLST, multilocus sequence typing; ND, not determined; NG-STAR, *Neisseria gonorrhoeae* Sequence Typing for Antimicrobial Resistance; O, ocular; PBP, penicillin binding protein; PEN, penicillin G (S ≥ 47 mm); R, resistant; S, sensitive; ST, sequence typing; TET, tetracycline (S ≥ 38 mm); U, urethral; V, vaginal. * Selected substitutions in *PenA* alleles. ** According to NG-STAR [20].

## Data Availability

The data presented in this study are available within the article.

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
