# Peer review of "The Molecular Genetic Epidemiology and Antimicrobial Resistance of Neisseria gonorrhoeae Strains Obtained from Clinical Isolates in Central Panama"

_microorganisms, 2023, doi:10.3390/microorganisms11102572_

Round 1

Reviewer 1 Report

Reviewer Comments:

Introduction:

The introduction presents a concise and enlightening synopsis of Neisseria gonorrhoeae, its prevalence, and the escalating issue of antimicrobial resistance (AMR). The study's context and significance have been thoroughly demonstrated. Nevertheless, the study fails to establish a clear research objective or formulate a hypothesis. Explicitly stating the purpose of the study and its contribution to the field would be beneficial.

Methods:

The portion pertaining to the study design and data analysis is presented in a coherent and useful manner. Nevertheless, it would be advantageous to provide further details regarding the sampling methodologies employed, the demographic characteristics of the patients, and the underlying reasoning behind the selection of antibiotics subjected to testing.

Results:

The presentation of findings is thorough and exhibits a clear and logical structure. The utilization of tables has proven to be efficacious. The assessment of the prevalence of antibiotic resistance among strains of N. gonorrhoeae in the research location holds significant importance. However, it is of utmost importance to analyze and interpret these data and engage in a comprehensive discussion regarding their implications within the framework of regional and worldwide trends in antimicrobial resistance (AMR). The examination of surveillance technologies (STs) and the analysis of resistance patterns contribute to the comprehensive understanding of the findings.

Discussion.

The discourse provides valuable information and presents the findings within the framework of comparative analysis with other locations; yet, there is potential for further expansion. It is advisable to offer a more comprehensive examination of the importance of the sequence types (STs) found in Panama and their possible implications for antimicrobial resistance (AMR). Furthermore, it is of great importance to examine the clinical ramifications associated with elevated levels of ciprofloxacin resistance and the potential influence it may exert on treatment protocols in the context of Panama.

Limitations.

The study acknowledges its limitations; yet, it is crucial to underscore the consequences of these constraints on the generalizability of the study and the accuracy of its conclusions. Furthermore, it would be beneficial to consider the limited sample size and potential biases in data collecting in order to bolster the study's credibility.

Minor

Author Response

REVIEWER 1

Comments and Suggestions for Authors

Reviewer Comments:
Introduction:
The introduction presents a concise and enlightening synopsis of Neisseria gonorrhoeae, its prevalence, and the escalating issue of antimicrobial resistance (AMR). The study's context and significance have been thoroughly demonstrated. Nevertheless, the study fails to establish a clear research objective or formulate a hypothesis. Explicitly stating the purpose of the study and its contribution to the field would be beneficial.

Response: Thank you for your comment and suggestions. We have added new content in the introduction and included what the reviewers kindly suggested.

Methods:

The portion pertaining to the study design and data analysis is presented in a coherent and useful manner. Nevertheless, it would be advantageous to provide further details regarding the sampling methodologies employed, the demographic characteristics of the patients, and the underlying reasoning behind the selection of antibiotics subjected to testing.

Response: Many thanks for your valuable comment.

A collection of Neisseria gonorrhoeae isolates collected from samples processed as part of routine clinical care during the years 2013-2018 at the Clinical Laboratory of the Luis Chicho Fabrerga Hospital was used for this study. Therefore, all isolates were collected as part of laboratory surveillance.  Data that were captured in the database as part of the laboratory surveillance were used to obtain the study variables.

We have added new content to the methods and included the kind suggestions of the reviewers.

Results:

The presentation of findings is thorough and exhibits a clear and logical structure. The utilization of tables has proven to be efficacious. The assessment of the prevalence of antibiotic resistance among strains of N. gonorrhoeae in the research location holds significant importance. However, it is of utmost importance to analyze and interpret these data and engage in a comprehensive discussion regarding their implications within the framework of regional and worldwide trends in antimicrobial resistance (AMR). The examination of surveillance technologies (STs) and the analysis of resistance patterns contribute to the comprehensive understanding of the findings.

Response: Thank you for this suggestion. We have now added to the discussion epidemiological data comparing the ST of the N. gonorrhoeae populations from countries in the Americas region.

Discussion.

The discourse provides valuable information and presents the findings within the framework of comparative analysis with other locations; yet, there is potential for further expansion. It is advisable to offer a more comprehensive examination of the importance of the sequence types (STs) found in Panama and their possible implications for antimicrobial resistance (AMR). Furthermore, it is of great importance to examine the clinical ramifications associated with elevated levels of ciprofloxacin resistance and the potential influence it may exert on treatment protocols in the context of Panama.

Response:  Many thanks for your valuable comment.  We have now added to the discussion epidemiological data comparing the STs of the N. gonorrhoeae populations of countries in the Americas region. The discussion section (lines 229-237) examines the clinical ramifications associated with elevated levels of ciprofloxacin resistance and the potential influence it may have on treatment protocol guidelines in the context of Panama.

Limitations.

The study acknowledges its limitations; yet, it is crucial to underscore the consequences of these constraints on the generalizability of the study and the accuracy of its conclusions. Furthermore, it would be beneficial to consider the limited sample size and potential biases in data collecting in order to bolster the study's credibility.

 Response:  Many thanks for your valuable comment. We have now written the phrase as: " limited sample size "

Reviewer 2 Report

The manuscript is overall polished and scientifically sound. The introduction is well-crafted and written in a plain, easy-to-read manner. The aim of the study is clearly stated. The method section includes details that are reproducible while being brief and concise. The results section contains simple data analysis from 30 isolates using both the traditional culture method and molecular detection method. The discussion makes an in-depth connection with literature that has been known while highlighting specifics in South America.

I have 2 minor comments.

1. figure1: included n=30 in the figure or figure legend to clarify the number inside the bars
2. There are discrepancies between antimicrobial susceptibility testing and molecular typing results. Discussion is needed.

Author Response

REVIEWER 2

Comments and Suggestions for Authors

The manuscript is overall polished and scientifically sound. The introduction is well-crafted and written in a plain, easy-to-read manner. The aim of the study is clearly stated. The method section includes details that are reproducible while being brief and concise. The results section contains simple data analysis from 30 isolates using both the traditional culture method and molecular detection method. The discussion makes an in-depth connection with literature that has been known while highlighting specifics in South America.

Reviewer Comments:

figure1: included n=30 in the figure or figure legend to clarify the number inside the bars

Response: Thank you for your comments and suggestions. We have added n=30 in figure 1

There are discrepancies between antimicrobial susceptibility testing and molecular typing results. Discussion is needed.

Response: Thank you for pointing out this error. We have already corrected the error in table 1 (isolated 23).

Reviewer 3 Report

This article presents findings on molecular typing and antimicrobial resistance in Neisseria gonorrhoeae in Panama, a region that has not been extensively researched. The topic is of great importance and relevance for a comprehensive understanding of the global situation and for local public health. Below are a few suggestions for improvement.

1) Firstly, please correct the antibiotic abbreviations throughout the text and in figures and tables to adhere to the generally accepted nomenclature: use "AZM" instead of "ATH" for Azithromycin. Penicillin should be referred to as "PEN," not "PG," and Tetracycline as "TET," not "T3O."

2) As for Fig.1, the Intermediate and Resistant categories should not be combined but represented separately with different colours: green for Sensitive, yellow for Intermediate, and red for Resistant.

3) Additionally, the number of Intermediate+Resistant for penicillin, tetracycline, and ciprofloxacin is incorrect. In Table 1, there are 30, but both the text and picture show only 29. Table 1 displays 25 isolates for TET, one less than the number indicated in the table header. However, the picture also shows 25 isolates. Similarly, the CIP section of the table indicates 12 isolates, but the text and picture show 13.

4) Regarding the ST MLST column, the abbreviation "ND" likely denotes "no data" rather than a new sequencing type.  

5) Please remove the eight in line 181, which might be a reference to a quotation and should be enclosed in square brackets if necessary.

5) Correction: Line 212 should read "PBP proteins" instead of "PPB proteins". 

6) Correction: A typo appears in line 232 where "ADR" should be corrected to "AMR", referring to antimicrobial resistance. 

7) To ensure a comprehensive analysis of epidemiological data, a comparison of neighbouring countries' Ng populations by MLST types could be useful.  For instance, according to Reimche et al. (Microbial Genomics 2023 https://doi.org/10.1099/mgen.0.001006), the dominant MLST types in the USA are 9363, 10314, 8143, and 1599. According to da Costa-Lourenço et al. (Infection, Genetics and Evolution 2018 https://doi.org/10.1016/j.meegid.2017.12.003), MLST types 1901 and 1588 are predominant in Rio de Janeiro, Brazil. Additionally, a genetic line with ST 8145 was discovered in Brazil. However, the most frequent MLST type in the authors' work, type 11516, was not detected in either Brazil or the USA. The observations imply that Ng in Panama has a localized nature and is distant from the types found in the USA, but closer to those in Brazil. It is recommended that the authors do a phylogenetic tree of their MLST types and those from other American countries.

Author Response

REVIEWER 3

Comments and Suggestions for Authors

Reviewer Comments:

This article presents findings on molecular typing and antimicrobial resistance in Neisseria gonorrhoeae in Panama, a region that has not been extensively researched. The topic is of great importance and relevance for a comprehensive understanding of the global situation and for local public health. Below are a few suggestions for improvement.

Firstly, please correct the antibiotic abbreviations throughout the text and in figures and tables to adhere to the generally accepted nomenclature: use "AZM" instead of "ATH" for Azithromycin. Penicillin should be referred to as "PEN," not "PG," and Tetracycline as "TET," not "T3O."

Response: Thanks for your valuable comments. We have now changed “ATH” to “AZM”, “PG” to “PEN” and “T3O” to “TET”.

As for Fig.1, the Intermediate and Resistant categories should not be combined but represented separately with different colours: green for Sensitive, yellow for Intermediate, and red for Resistant.

Response: Thank you for your comments and suggestions. We have modified figure 1 and added your suggestions.

Additionally, the number of Intermediate+Resistant for penicillin, tetracycline, and ciprofloxacin is incorrect. In Table 1, there are 30, but both the text and picture show only 29. Table 1 displays 25 isolates for TET, one less than the number indicated in the table header. However, the picture also shows 25 isolates. Similarly, the CIP section of the table indicates 12 isolates, but the text and picture show 13.

Response: Thank you for pointing out this error. We have already corrected the error in table 1.

Regarding the ST MLST column, the abbreviation "ND" likely denotes "no data" rather than a new sequencing type. 

Response: We have now added in the legend of the tables 1 explanation for ND: not determined

Please remove the eight in line 181, which might be a reference to a quotation and should be enclosed in square brackets if necessary.

Response: Thank you for pointing out this error. We have bracketed the number 8 as a reference to a quotation.

Correction: Line 212 should read "PBP proteins" instead of "PPB proteins".

Response: Many thanks to the reviewer for the valuable comment. We have now replaced "PPB proteins" with "PBP proteins"

Correction: A typo appears in line 232 where "ADR" should be corrected to "AMR", referring to antimicrobial resistance. 

Response: Thank you for pointing out this error. We have now changed "ADR” to "AMR".

To ensure a comprehensive analysis of epidemiological data, a comparison of neighbouring countries' Ng populations by MLST types could be useful.  For instance, according to Reimche et al. (Microbial Genomics 2023 https://doi.org/10.1099/mgen.0.001006), the dominant MLST types in the USA are 9363, 10314, 8143, and 1599. According to da Costa-Lourenço et al. (Infection, Genetics and Evolution 2018 https://doi.org/10.1016/j.meegid.2017.12.003), MLST types 1901 and 1588 are predominant in Rio de Janeiro, Brazil. Additionally, a genetic line with ST 8145 was discovered in Brazil. However, the most frequent MLST type in the authors' work, type 11516, was not detected in either Brazil or the USA. The observations imply that Ng in Panama has a localized nature and is distant from the types found in the USA, but closer to those in Brazil. It is recommended that the authors do a phylogenetic tree of their MLST types and those from other American countries.

Response: Thank you for your comments and suggestions. We have added new content in the discussion, as suggested by the reviewer. In addition, we have included the associated references.

See Phylogenetic tree attached 

  Neisseria gonorrhoeae: geoBurst

In  Figure 1 we observe the minimum spanning tree showing the relationship between the most prevalent sequence types (STs) of Neisseria gonorrhoeae constructed using geoBurst MLST with the Phyloviz 2.0 program. Each circle reflects one ST, and the color reflects the geographic information or area of the isolate. The value between each circle indicates the gene difference between the STs. A Minimum spanning tree was performed to establish the relationship of the different types of sequences (STs) obtained from N. gonorrhoeae from the present work in relation to da Costa-Lourenço et al., 2018; Gianecini et al., 2021; Reimche et al., 2023. The MLST geoBurst was constructed with Phyloviz 2.0 (Francisco et al., 2012).

The most prevalent Neisseria gonorrhoeae strains of ST in Panama are phylogenetically closer to the strains identified in Brazil.
